# US women screen at low rates for both cervical and colorectal cancers than a single cancer: a cross-sectional population-based observational study

Diane M Harper[1,2,3]*, Melissa Plegue[1], Masahito Jimbo[1,4], Sherri Sheinfeld Gorin[1], Ananda Sen[1,5]

[1]Department of Family Medicine, University of Michigan, Ann Arbor, United States; [2]Department of Obstetrics and Gynecology, University of Michigan, Ann Arbor, United States; [3]Department of Women's and Gender Studies, University of Michigan, Ann Arbor, United States; [4]Department of Family and Community Medicine, University of Illinois, Chicago, United States; [5]Department of Biostatistics, University of Michigan, Ann Arbor, United States

*For correspondence:
harperdi@med.umich.edu

## Abstract

**Background:** Using screen counts, women 50–64 years old have lower cancer screening rates for cervical and colorectal cancers (CRC) than all other age ranges. This paper aims to present woman-centric cervical cancer and CRC screenings to determine the predictor of being up-to-date for both.

**Methods:** We used the Behavioral Risk Factor Surveillance System (BRFSS), an annual survey to guide health policy in the United States, to explore the up-to-date status of dual cervical cancer and CRC screening for women 50–64 years old. We categorized women into four mutually exclusive categories: up-to-date for dual-screening, each single screen, or neither screen. We used multinomial multivariate regression modeling to evaluate the predictors of each category.

**Results:** Among women ages 50–64 years old, dual-screening was reported for 58.2% (57.1–59.4), cervical cancer screening alone (27.1% (26.0–28.2)), CRC screening alone (5.4% (4.9–5.9)), and neither screen (9.3% (8.7–9.9)). Age, race, education, income, and chronic health conditions were significantly associated with dual-screening compared to neither screen. Hispanic women compared to non-Hispanic White women were more likely to be up-to-date with cervical cancer screening than dual-screening (adjusted odds ratio [aOR] = *1.39 (1.10, 1.77)*). Compared to younger women, those 60–64 years are significantly more likely to be up-to-date with CRC screening than dual-screening (aOR = *1.75 (1.30, 2.35)*).

**Conclusions:** Screening received by each woman shows a much lower rate of dual-screening than prior single cancer screening rates. Addressing dual-screening strategies rather than single cancer screening programs for women 50–64 years may increase both cancer screening rates.

**Funding:** This work was supported by NIH through the Michigan Institute for Clinical and61 Health Research UL1TR002240 and by NCI through The University of Michigan Rogel Cancer62 Center P30CA046592 grants.

## Editor's evaluation

This work presents a different US perspective for cancer screenings among the least screened women, those 50-64 years old. Using the Behavioral Risk Factor Surveillance System and the two cancer screenings (cervical and colorectal) done by her primary care physician, the findings of this

work show that women-centric screening is lower than single cancer screenings. There is a large gap in health systems processes to meet the cancer screening needs of each woman.

## Introduction

One of the primary goals of Healthy People 2030 is to improve cancer screening behaviors with evidence-based screening and prevention strategies (HP2030-CR02 *Office of Disease Prevention and Health Promotion, 2020*). On the population level, the Center for Disease Control and Prevention (CDC) issues the Behavioral Risk Factor Surveillance System (BRFSS) survey which collects data on cancer screenings by cancer type, screening method, and by the time from their last screening (*Behavioral Risk Factor Surveillance System, 2019*). BRFSS is the United States' premier database used to report single cancer screening rates, such as cervical cancer or colorectal cancer (CRC). Never has the perspective of woman-centered screening been published reporting the patterns of screening for both cervical cancer and CRC (*dual-screening*) among women 50–65 years old.

CRC screening has never approached the Healthy People goals, but up-to-date screening has shown a steady increase, regardless of screening method (*National Cancer Institute Cancer Trends Progress Reports, 2020a*). Most recently, however, BRFSS showed a significant gap in CRC screening among women 50–64 years of age, with self-reported rates as low as 50% for the youngest age-appropriate women (*Joseph et al., 2020*). At this time, among women 50–64 years old, the incidence of CRC is 60/100,000 women, and the mortality is 20/100,000 women (*Seigel, 2020*). Unlike CRC screening, cervical cancer screening prevalence peaked at the turn of the 21st century and has been declining year over year (*National Cancer Institute Cancer Trends Progress Reports, 2020b*, summary table). Like CRC screening, the lowest cervical cancer screening rates are among women who were 50–64 years old (*Harper et al., 2020*). Cervical cancer incidence and mortality are reported in 5-year time frames: among women 50–54 years old, the incidence is 12.5/100,000; 55–59 years old: 12.1/100,000; 60–64 years old: 10.8/100,000. The mortality for the same age groups is 4.6/100,000, 4.6/100,000, and 4.3/100,000 (*US Cancer Statistics Working Group, 2021US Cancer Statistics Working Group, 2021US Cancer Statistics Working Group, 2021*).

Both cervical cancer and CRC generally do not have symptoms until a very late stage when cure is not possible (*World Health Organization, 2017*), making early screening, even self-screening (*El Khoury et al., 2021*; *Bakr et al., 2020*; *Jaklevic, 2020*), a critical health imperative.

This paper aims to present cervical cancer and CRC screenings per woman 50–64 years old and determine the predictors of being up-to-date in <u>both</u> cervical cancer and CRC screening (*dual-screened*) compared to each single screen.

## Materials and methods

### Survey data

BRFSS is the premier US health-related telephone survey that collects state-level data about US residents regarding their health-related risk behaviors as well as their completion of clinical preventive services. BRFSS is an annual, state-based, cross-sectional telephone survey that US state health departments conduct monthly using landline telephones and cellular telephones using a standardized questionnaire developed and sponsored by the CDC. CRC screening is defined as using a home kit for blood stool tests including fecal occult blood test (FOBT) or fecal immunochemical test (FIT) or office-based procedures including sigmoidoscopy or colonoscopy. The interval options since the last screening include (1) within the past year (anytime less than 12 months ago), (2) within the past 2 years (1 year but less than 2 years ago), (3) within the past 3 years (2 years but less than 3 years ago), (4) within the past 5 years (3 years but less than 5 years ago), and (5) 5 or more years ago. The responses for cervical cancer screening are based on office-based testing for HPV, Pap test, or both. The interval options since the last screening were identical to those for CRC screening. The BRFSS dataset available for our analysis was 2018.

### Screening outcomes

Cervical cancer screening was considered up-to-date if the respondent reported a Pap within the past 3 years, an HPV test within the past 5 years, or a Pap and HPV test within the past 5 years (*Curry et al.,*

**eLife digest** Routine screenings for cervical and colorectal cancers save lives by detecting cancers at an early stage when they are more treatable and more likley to cure. Most cancer screening in the United States is focused on single cancer screening programs, often held at community health fairs, pop-up screening vans and other settings, without coordination with the individuals' primary care doctors.

This is problematic because the primary care physician cannot counsel if the results are abnormal and advise when the next routine screen is appropriate. This leads to gaps in women not being informed that they are due for routine screening and gaps to act on any abnormal screening results. This is especially problematic for women aged 50 to 64, who are less likely to screen for either cancer alone compared to other age groups.

Currently, 86% of women in the United States are up to date with cervical cancer screening, and 64% are up to date with colorectal cancer screening. However, it is not clear how many women in this age group receive both screens, compared to a single screen or neither screen.

Harper et al. analyzed data from over 40,000 women aged 50 to 64, collected in a United States health survey in 2018. This study revealed that only 59% of the women reported being up to date with cervical and colorectal cancer screenings. Compared to women who did not screen at all, women completing both screens were more educated, had higher incomes, and were more likely to have other chronic conditions such as arthritis, diabetes, depression and other cancers.

These findings reveal that the number of women aged 50 to 64 in the United States, who are up to date with both cancer screenings, is still well below national targets. Harper et al. propose that shifting towards a women-centric focus, with primary care physicians or health care systems responsible for managing screening efforts, could decrease cancer incidence and mortality. In future, self-test kits for both cancers should help encourage more women to have both screens in a comfortable environment. This change in focus will also allow primary care physicians to notify women at appropriate intervals to attend routine screening and immediate follow-ups in the case of abnormal results.

*2018*). We included the 5-year screening interval in our outcome measure of cervical cancer screening. For CRC screening, up-to-date screening included FOBT or FIT within the past year, a sigmoidoscopy within the last 5 years, or a colonoscopy within the past 10 years, in accordance with the USPSTF guidelines (*Force et al., 2016*; *Force et al., 2021*). To examine up-to-date screening for both cancers, we grouped women into four mutually exclusive screening categories: those who were up-to-date for both screens (dual-screening), those up-to-date with a single screen (cervical cancer screen only or CRC screen only), and those who denied completing either screen.

## Survey respondents

We included women aged 50–64 years with no reported history of uterine or cervical cancer and who had not had a hysterectomy. We also excluded those with reported colon or rectal cancers and those who did not answer both the cervical cancer and CRC screening questions.

## Statistical analysis

Data were summarized using proportions, unweighted and weighted, with 95% confidence intervals. The dataset was analyzed using weighted sampling methods to ensure valid inferences from the responding sample to the general population, correcting for nonresponse and noncoverage biases (*StataCorp, 2017*; *BRFSS sample weights, 2018*). Composite up-to-date screening prevalences were summarized across all screening methods for each cancer screen. For instance, a woman was up-to-date with CRC screening if she had reported a FIT or FOBT test within the past 1 year OR a sigmoidoscopy within the past 5 years OR a colonoscopy within the past 10 years. Likewise, a woman was up-to-date with cervical cancer screening if she had reported a Pap test within the past 3 years OR a combined Pap and HPV test within the past 5 years OR an HPV test within the past 5 years. Women with complete screening status were included in the analysis.

In order to reduce potential bias due to missingness, we employed inverse propensity weighting (*Rosenbaum, 1987*; *Office of Disease Prevention and Health Promotion, 2019*). Propensity weight

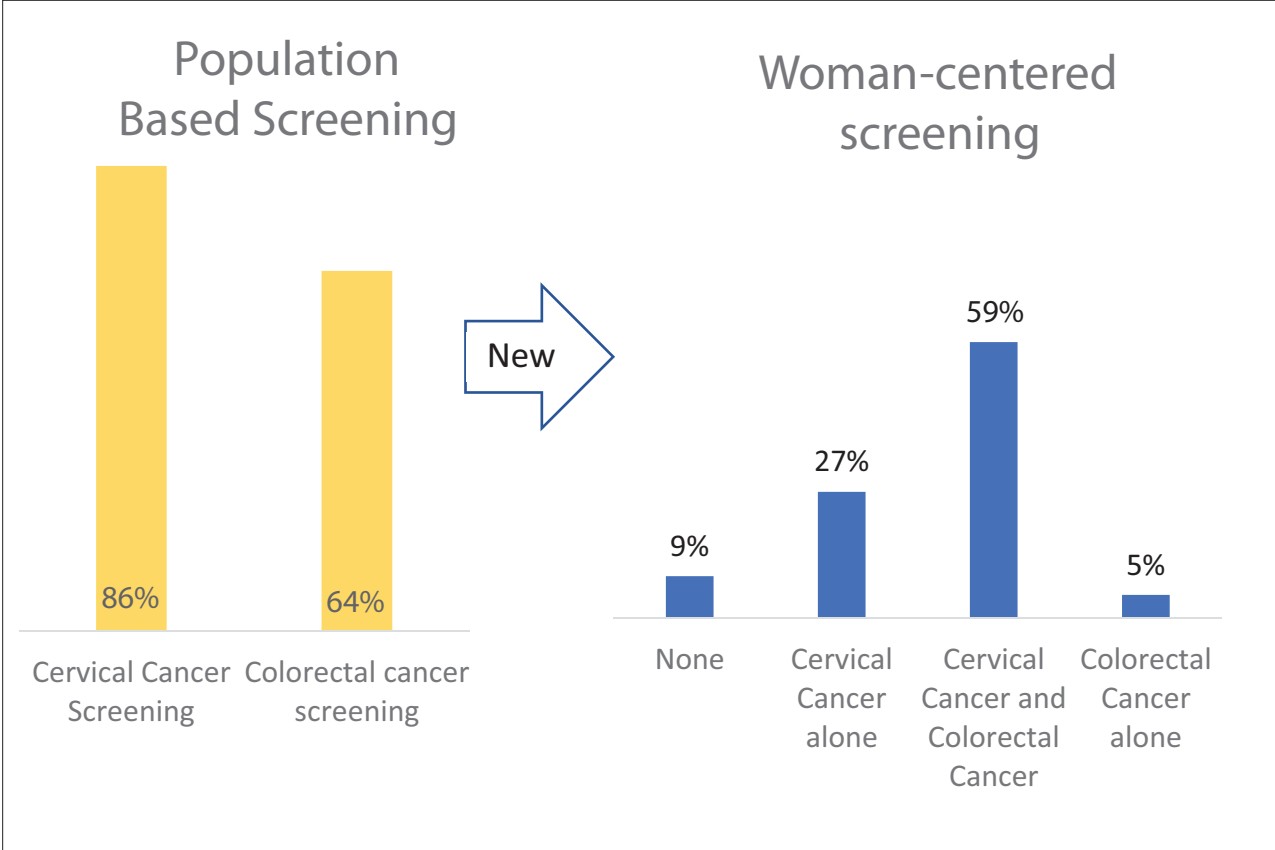

**Figure 1.** Graphical abstract. Screening rates differ by calculation approach. A patient-centered approach considers the total number of screens each woman has, whereas a population-centered system documents each single test completion. Our data show much lower women-centered cervical cancer and colorectal cancer (CRC) screening rates than a population-based screening approach. This difference is especially relevant as women may be able to do these two cancer screenings at home by themselves.

was estimated as the predicted probability of nonmissingness based on a logistic regression with age, race, marital status, education, income, employment status, and insurance status as covariates all of which significantly predicted missingness. The inverse of the propensity weights was further multiplied with the BRFSS survey weights to construct updated weights, which were subsequently used in all models for the complete cases.

Multinomial logistic regression was used to evaluate the associations with the predictors of screening outcomes. Predictors of interest were age, race, education, income, marital status, occupational status, geolocation, and the presence of any chronic health condition as found to be pertinent in past work (*Harper et al., 2020*). All tests of statistical significance used a p-value threshold of 0.05. This retrospective study was exempt from the IRB.

## Results

### Main results

There were 67,473 women identified between the ages of 50–64 years from which 40,511 met the inclusion and exclusion criteria and reported complete screening data for CRC and cervical cancers in the BRFSS 2018 survey (*Figure 1*). Weighted prevalences are presented in *Table 1*, where the largest proportion of women were of the White race, married, attended at least some college, lived in an urban area, had an income of over $50,000, and were employed.

*Table 2* presents mutually exclusive up-to-date screening categories by screening modality by the woman. The weighted prevalence for women with dual CRC and cervical cancer screening was 58.2% (57.1%, 59.4%), *n* = 24,678, for women with neither screen was 9.3% (8.7%, 9.9%), *n* = 3560, for women with only cervical cancer screening was 27.1% (26.0%, 28.2%), *n* = 10,051, and for women with

**Table 1.** Demographic characteristics of the study population.

| | Weighted % | Eligible for analysis (n = 40,511) |
|---|---|---|
| **Age, n (%)** | | |
| 50–54 | 35.8 | 12,165 (30.0) |
| 55–59 | 31.3 | 13,757 (34.0) |
| 60–64 | 32.9 | 14,589 (36.0) |
| **Race, n (%)** | | |
| NH White | 67.4 | 31,340 (77.4) |
| NH Black | 10.6 | 3401 (98.4) |
| NH Other/Multiracial | 6.9 | 2484 (6.1) |
| Hispanic | 13.5 | 2836 (7.0) |
| Unknown/Refused | 1.5 | 450 (1.1) |
| **Marital, n (%)** | | |
| Married/partnered | 63.4 | 25,022 (61.8) |
| Unmarried* | 35.9 | 15,335 (37.9) |
| Unknown/refused | 0.7 | 154 (0.4) |
| **Education, n (%)** | | |
| Less than high school | 11.6 | 2293 (5.7) |
| High school | 25.1 | 9447 (23.3) |
| Attended college/Tech school | 31.1 | 11,066 (27.3) |
| Graduated college/ Tech school | 32.1 | 17,671 (43.6) |
| Refused/don't know | 0.2 | 34 (0.1) |
| **Location, n (%)** | | |
| Urban | 92.7 | 33,500 (82.7) |
| Rural | 6.2 | 6292 (15.5) |
| Unknown/missing | 1.1 | 719 (1.8) |
| **Income, n (%)** | | |
| <50,000 | 36.5 | 14,870 (36.7) |
| 50,000+ | 46.4 | 20,295 (50.1) |
| Don't know/refused | 17.1 | 5346 (13.2) |
| **Employment status†, n (%)** | | |
| Employed/self-employed | 59.5 | 25,289 (62.4) |
| Unemployed, looking | 5.4 | 1866 (4.6) |
| Unemployed, not looking | 21.3 | 8392 (20.7) |
| Unable to work | 12.7 | 4763 (11.8) |
| Refused | 1.2 | 201 (0.5) |

*Unmarried includes divorced, widowed, separated, and never married.

†'Unemployed, not looking' includes 'Homemaker, Student, and Retired', 'Unemployed, looking' includes out of work over a year and out of work less than a year.

only CRC screening was [5.4% (4.9%, 5.9%)], n = 2222. The tests used by the majority of women for CRC and cervical cancer screening, respectively, were colonoscopy and a Pap test (54.6% (53.2%, 55.9%)). Women who chose only one screen chose cervical cancer screening five times more often than CRC screening (27.1% vs. 5.4%). By traditional screening calculations, cervical cancer screening was 86% (85.4–86.1) and CRC screening was 66% (65.9–66.8) (see *Figure 1*).

The **descriptive analyses** (*Table 3*) showed differences among women across the four mutually exclusive groups. **Age** by 5-year intervals showed that the youngest group was least screened for both CRC and cervical cancer (dual-screening) (47.4%) as well as CRC screening alone (3.1%). However, this age group had the highest screening for cervical cancer alone (40.0%). Among **races**, Hispanic women had the lowest dual-screening (46.5%), the lowest screening for CRC alone (5.2%), and yet had the highest screening for cervical cancer alone (39.4%).

Among women of all **education** levels, those with the least education had the lowest dual-screening (41.0%), but the most for a single cancer (cervical (36.8%); CRC alone (6.9%)). In addition, the least educated women had the highest percentage of not participating in either screen (15.3%). Conversely, among all **income** levels, those with the highest income reported the greatest dual-screening (65.7%) and the lowest single cancer screening (cervical: 24.5%; CRC: 4.3%).

Among **marital** categories, never-married women reported the lowest dual-screening (53.7%); and the highest for the single screens (cervical: 28.2%; CRC 6.0%). Unemployed women (homemaker, student, or retired) among all **occupational** categories reported the highest dual-screening (62.5%), while those living in rural areas reported the lowest dual-screening (52.0%). 60.7% of women with chronic conditions completed dual-screening.

Tables of the univariate multinomial regression models are discussed as results in *Appendix 1— table 1*.

## Multivariate multinomial logistic regression

Multivariate multinomial logistic regression was used to evaluate screening outcomes adjusted for the significant model matched variables of age, race, education, income, and some chronic conditions. Wald tests were used to test the inclusion of the variable in the overall model where marital

**Table 2.** Up-to-date screening status by woman by screening modality.

| | Dual screened, *n* = 24,678; 58.2% (57.1, 59.4) | | | |
| --- | --- | --- | --- | --- |
| | Pap only, within last 3 years | HPV only, within last 5 years | Pap and HPV, within last 5 years | Current for colorectal only, *n* = 2222 5.4% (4.9, 5.9) |
| Blood stool only, last year | 5.0 (4.3, 5.8) | 0.1 (0.001, 0.2) | 3.0 (2.5, 3.6) | 10.0 (7.7, 13.1) |
| Sigmoidoscopy only, last 5 years | 0.6 (0.4, 0.8) | <0.1% | 0.4 (0.3, 0.5) | 1.2 (0.6, 2.3) |
| Colonoscopy only, last 10 years | 54.6 (53.2, 55.9) | 1.9 (1.4, 2.5) | 27.2 (26.0, 28.3) | 83.3 (79.4, 86.6) |
| Two CRC tests | 3.8 (3.4, 4.3) | <0.1% | 3.3 (2.8, 4.0) | 5.5 (3.4, 8.6) |
| Current for cervical cancer only, *n* = 10,051 27.1% (26.0, 28.2) | 66.5 (64.2, 68.6) | 2.9 (2.2, 3.7) | 30.7 (28.6, 32.8) | 100% |

Not current for either, *n* = 3560; 9.3% (8.7, 9.9)

Women receiving only one of the two screens

CRC: The left-hand column lists the types of colorectal cancer screening that a woman could report. The far right-hand column presents the weighted screening proportions (95% CI) of women with only CRC screening by that modality. For instance, 10.0% of the 5.4% of women who were only current for CRC indicated they had been screened with a blood stool test within the last year but had no other CRC screen and no other cervical cancer screen. Likewise, 83.3% of the 5.4% of women who were only current on CRC screening (but had no other CRC screen and no other cervical cancer screen) indicated that they had a colonoscopy within the past 10 years.

Cervical: The second to bottom row indicates the weighted screening proportions (95% CI) of women who only were up- to -date with cervical cancer screening by the columnar headings for types of cervical cancer screening modalities. For instance, 66.5% of the 27.1% of women who only had cervical cancer screening received a Pap test within the past 3 years. Likewise, 30.7% of the 27.1% of women who only had cervical cancer screening had received co-testing within the past 5 years.

Women who had both screens by modality

The top row indicates that 58.2% of all women were up-to-date with both cervical cancer and colorectal cancer screening (dual-screening). The inner 12 cells indicate the weighted percentage of women with dual-screening by modality. For instance, 54.6% of the 58.2% of women having both screens up-to-date did so with a colonoscopy and a Pap test. Likewise, 27.4% of 58.7% of women having both screens up-to-date did so with colonoscopy and cotesting.

Women not up-to-date with either screen

The bottom row indicates that 9.3% of the analytic cohort received neither screen because of underscreening or no screening.

CI, confidence interval; CRC, colorectal cancer.

status (p = 0.202), current employment (p = 0.189), geographic location (p = 0.08), cardiac comorbidities (p = 0.12), kidney comorbidities (p = 0.11), and lung comorbidities (p = 0.64) were no longer significant and, hence, were dropped from the model (*Table 4*).

## Screening compared to neither screening

1a. Women who graduated from college compared to less than high school graduation (adjusted odds ratio aOR = 3.35 (2.33, 4.81)), who had the highest income level (aOR = 3.32 (2.64, 4.18)) or had experienced prior cancer (aOR = 2.90 (2.09, 4.05)) had the highest adjusted odds of **completing dual-screening** compared to no screening. In addition, women who were older aOR = 1.35 (1.09, 1.67) and aOR = **1.4**6 (1.18, 1.79) for 55–59 and 60–64 years, respectively, compared to 50–54 years old, Black aOR = 2.25 (1.62, 3.13) or Hispanic (aOR = 1.68 (1.25, 2.25)) compared to White, and past history of arthritis aOR = 1.83 (1.54, 2.18), diabetes aOR = 1.34 (1.06, 1.69), or depression aOR = 1.35 (1.11, 1.63) were also significantly more likely to be up-to-date for **both screens** compared to neither screen.

1b. Black and Hispanic women compared to White had the highest adjusted odds of completing **just the cervical cancer screening** compared to neither screening aOR = 1.73 (1.22, 2.44) and aOR = 2.34 (1.72, 3.18), respectively. In addition, women with the highest income level (aOR = 2.19 (1.71, 2.80)), who graduated from college compared to those with less than high school graduation (aOR = 2.02 (1.39, 2.94)) or had prior cancer (aOR = 2.28 (1.57, 3.31)) were also more than twice as likely to have only cervical cancer screening compared to having no screening. Finally, younger women, and those with a past history of arthritis (aOR = 1.28 (1.06, 1.55)), or diabetes (aOR = 1.34 (1.03, 1.75)) were more likely to screen for **cervical cancer alone** than neither screening.

1c. Women with prior cancer had the highest adjusted odds of completing **just the CRC screen** compared to having neither screening (aOR = 3.08 (1.94, 4.91)). The oldest women (aOR = 2.55 (1.81, 3.58) for 60–64 years old, aOR = 1.76 (1.25, 2.47) for 55–59 years old), those with the highest income level (aOR = 2.13 (1.54, 2.96)) or those with a past history of arthritis (aOR = 2.17 (1.65, 2.87)) were more than two times more likely to have CRC screening alone compared to neither screening. Finally,

**Table 3.** Sociodemographic descriptors by screening category.

| Total | Dual-screening | | | Cervical cancer screening alone | | | CRC screening alone | | | Neither screening | | |
|---|---|---|---|---|---|---|---|---|---|---|---|---|
| n = 40,511 | N = 24,678 | | | N = 10,051 | | | N = 2222 | | | N = 3560 | | |
| | N | Unweighted row % | Weighted row % | N | Unweighted row % | Weighted row % | N | Unweighted row % | Weighted row % | N | Unweighted row % | Weighted row % |
| **Age, (n, row %)** | | | | | | | | | | | | |
| 50–54 | 6121 | 50.3 | 47.4 | 4603 | 37.8 | 40.0 | 355 | 2.9 | 3.1 | 1086 | 8.9 | 9.5 |
| 55–59 | 8864 | 64.4 | 62.5 | 2893 | 21.0 | 22.7 | 778 | 5.7 | 5.2 | 1222 | 8.9 | 9.6 |
| 60–64 | 9693 | 66.4 | 66.0 | 2555 | 17.5 | 17.3 | 1089 | 7.5 | 7.9 | 1252 | 8.6 | 8.8 |
| **Race (n, row %)** | | | | | | | | | | | | |
| White | 19,507 | 62.2 | 60.3 | 7345 | 23.4 | 24.8 | 1799 | 5.7 | 5.6 | 2689 | 8.6 | 9.3 |
| Black | 2198 | 64.6 | 60.7 | 856 | 25.2 | 26.6 | 137 | 4.0 | 5.4 | 210 | 6.2 | 7.4 |
| Other* | 1335 | 53.7 | 59.2 | 706 | 28.4 | 25.2 | 135 | 5.4 | 3.0 | 308 | 12.4 | 12.6 |
| Hispanic | 1381 | 48.7 | 46.5 | 1025 | 36.1 | 39.4 | 126 | 4.4 | 5.2 | 304 | 10.7 | 8.8 |
| **Education (n, row %)** | | | | | | | | | | | | |
| Less than high school graduate | 965 | 42.1 | 41.0 | 775 | 33.8 | 36.8 | 152 | 6.6 | 6.9 | 401 | 17.5 | 15.3 |
| High school graduate | 5109 | 54.1 | 52.9 | 2574 | 27.3 | 27.7 | 583 | 6.2 | 6.5 | 1181 | 12.5 | 12.8 |
| Attended college/Tech school | 6587 | 59.5 | 60.0 | 2820 | 25.5 | 26.2 | 679 | 6.1 | 5.4 | 989 | 8.9 | 8.4 |
| Graduated college/Tech school | 11,994 | 67.9 | 67.0 | 3876 | 21.9 | 24.2 | 816 | 4.6 | 3.8 | 985 | 5.6 | 5.1 |
| **Income (n, row %)** | | | | | | | | | | | | |
| <$50K | 7779 | 52.3 | 49.7 | 4127 | 27.8 | 29.8 | 1021 | 6.9 | 6.7 | 1943 | 13.1 | 13.8 |
| ≥$50K | 13,711 | 67.6 | 65.7 | 4562 | 22.5 | 24.5 | 930 | 4.6 | 4.3 | 1092 | 5.4 | 5.5 |
| **Marital status (n, row %)** | | | | | | | | | | | | |
| Married or partnered | 15,966 | 63.8 | 61.0 | 6014 | 24.0 | 26.3 | 1230 | 4.9 | 5.0 | 1812 | 7.2 | 7.7 |
| Single* | 8630 | 56.3 | 53.7 | 3995 | 26.1 | 28.2 | 979 | 6.4 | 6.0 | 1731 | 11.3 | 12.1 |
| **Occupational status (n, row %)** | | | | | | | | | | | | |

*Table 3 continued on next page*

*Table 3 continued*

| Total | Dual-screening | | | Cervical cancer screening alone | | | CRC screening alone | | | Neither screening | | |
|---|---|---|---|---|---|---|---|---|---|---|---|---|
| | n | % | % | n | % | % | n | % | % | n | % | % |
| Employed | 15,613 | 61.7 | 58.6 | 6566 | 26.0 | 28.6 | 1126 | 4.4 | 4.5 | 1984 | 7.9 | 8.4 |
| Unemployed, looking† | 975 | 52.3 | 52.3 | 533 | 28.8 | 28.6 | 111 | 6.0 | 5.6 | 247 | 13.2 | 13.4 |
| Unemployed, not looking† | 5327 | 63.5 | 62.5 | 1758 | 21.0 | 23.3 | 543 | 6.5 | 5.8 | 764 | 9.1 | 9.0 |
| Unable to work | 2662 | 55.9 | 53.5 | 1132 | 23.8 | 25.6 | 435 | 9.1 | 9.2 | 534 | 11.2 | 11.7 |
| **Location (n, row %)** | | | | | | | | | | | | |
| Urban | 20,903 | 62.4 | 58.9 | 8105 | 24.2 | 26.8 | 1762 | 5.3 | 5.3 | 2730 | 8.2 | 9.0 |
| Rural | 3479 | 55.3 | 52.0 | 1662 | 26.4 | 28.4 | 431 | 6.9 | 7.2 | 720 | 11.4 | 12.4 |
| **Chronic conditions (n, row %)** | | | | | | | | | | | | |
| ANY | 16,168 | 63.2 | 60.7 | 5724 | 22.4 | 24.5 | 1631 | 6.4 | 6.4 | 2052 | 8.0 | 8.4 |
| Cardiac | 1117 | 56.5 | 55.1 | 457 | 23.1 | 23.5 | 176 | 8.9 | 8.4 | 228 | 11.5 | 13.1 |
| Stroke | 745 | 58.6 | 60.6 | 293 | 23.1 | 24.2 | 101 | 8.0 | 5.2 | 132 | 10.4 | 9.9 |
| Lung | 4039 | 60.2 | 58.5 | 1539 | 22.9 | 25.6 | 479 | 7.1 | 6.3 | 655 | 9.8 | 9.6 |
| Arthritis | 9799 | 64.3 | 62.9 | 3237 | 21.3 | 22.3 | 1049 | 6.9 | 7.2 | 1148 | 7.5 | 7.6 |
| Kidney | 751 | 62.3 | 63.2 | 242 | 20.1 | 19.5 | 96 | 8.0 | 5.2 | 117 | 9.7 | 12.1 |
| Diabetes | 2988 | 58.6 | 54.8 | 1233 | 24.2 | 28.9 | 413 | 8.1 | 7.6 | 468 | 9.2 | 8.7 |
| Depression | 5987 | 62.2 | 60.2 | 2133 | 22.2 | 24.5 | 695 | 7.2 | 6.7 | 805 | 8.4 | 8.5 |
| Skin cancer | 2182 | 70.4 | 68.4 | 540 | 17.4 | 18.7 | 189 | 6.1 | 5.1 | 190 | 6.1 | 6.1 |
| Other cancer | 2282 | 69.3 | 67.2 | 648 | 19.7 | 22.8 | 209 | 6.3 | 6.3 | 156 | 4.7 | 3.7 |

CRC, colorectal cancer.

*Single includes divorced, widowed, separated, and never married.

†'Unemployed, not looking' includes 'Homemaker, Student, and Retired', 'Unemployed, looking' includes out of work over a year and out of work less than a year.

**Table 4.** Multinomial multivariate regression.

| N = 42,701 Weighted n = 19,849,774 | Referent outcome: neither screen | | | Referent outcome: dual screens | | Referent outcome: cervical only |
|---|---|---|---|---|---|---|
| | Dual screen aOR (95% CI) | Cervical only aOR (95% CI) | CRC only aOR (95% CI) | Cervical only aOR (95% CI) | CRC only aOR (95% CI) | CRC only aOR (95% CI) |
| **Age** | | | | | | |
| 50–54 | Ref | Ref | Ref | Ref | Ref | Ref |
| 55–59 | 1.35 (1.09, 1.67) | 0.60 (0.48, 0.76) | 1.76 (1.25, 2.47) | 0.44 (0.39, 0.51) | 1.30 (0.98, 1.74) | 2.93 (2.17, 3.96) |
| 60–64 | 1.46 (1.18, 1.79) | 0.45 (0.36, 0.56) | 2.55 (1.81, 3.58) | 0.31 (0.27, 0.36) | 1.75 (1.30, 2.35) | 5.63 (4.14, 7.66) |
| **Race** | | | | | | |
| White | Ref | Ref | Ref | Ref | Ref | Ref |
| Black | 2.25 (1.62, 3.13) | 1.73 (1.22, 2.44) | 1.58 (0.95, 2.61) | 0.77 (0.64, 0.92) | 0.70 (0.46, 1.07) | 0.91 (0.59, 1.41) |
| Other* | 0.72 (0.46, 1.12) | 0.76 (0.47, 1.21) | 0.37 (0.21, 0.64) | 1.05 (0.74, 1.50) | 0.51 (0.33, 0.80) | 0.49 (0.30, 0.80) |
| Hispanic | 1.68 (1.25, 2.25) | 2.34 (1.72, 3.18) | 1.42 (0.77, 2.63) | 1.39 (1.10, 1.77) | 0.84 (0.48, 1.50) | 0.61 (0.34, 1.10) |
| **Education** | | | | | | |
| Less than high school | Ref | Ref | Ref | Ref | Ref | Ref |
| High school graduate | 1.37 (1.02, 1.83) | 1.08 (0.80, 1.46) | 1.14 (0.65, 1.98) | 0.78 (0.61, 1.01) | 0.83 (0.49, 1.39) | 1.06 (0.62, 1.80) |
| Attended college/Tech school | 1.98 (1.46, 2.69) | 1.47 (1.07, 2.00) | 1.30 (0.76, 2.22) | 0.74 (0.58, 0.95) | 0.65 (0.40, 1.07) | 0.88 (0.53, 1.46) |
| Graduated college/Tech school | 3.35 (2.33, 4.81) | 2.02 (1.39, 2.94) | 1.66 (0.94, 2.93) | 0.60 (0.46, 0.79) | 0.50 (0.31, 0.81) | 0.82 (0.49, 1.36) |
| **Income** | | | | | | |
| <$50K | Ref | Ref | Ref | Ref | Ref | Ref |
| ≥$50K | 3.32 (2.64, 4.18) | 2.19 (1.71, 2.80) | 2.13 (1.54, 2.96) | 0.66 (0.58, 0.75) | 0.64 (0.49, 0.84) | 0.98 (0.74, 1.29) |
| **Chronic conditions†** | | | | | | |
| Arthritis | 1.83 (1.54, 2.18) | 1.28 (1.06, 1.55) | 2.17 (1.65, 2.87) | 0.70 (0.61, 0.80) | 1.19 (0.94, 1.50) | 1.70 (1.31, 2.19) |
| Diabetes | 1.34 (1.06, 1.69) | 1.34 (1.03, 1.75) | 1.79 (1.25, 2.55) | 1.00 (0.83, 1.22) | 1.33 (0.98, 1.81) | 1.33 (0.95, 1.86) |

*Table 4 continued on next page*

*Table 4 continued*

| N = 42,701 Weighted n = 19,849,774 | Referent outcome: neither screen | | | Referent outcome: dual screens | | Referent outcome: cervical only |
|---|---|---|---|---|---|---|
| | Dual screen | Cervical only | CRC only | Cervical only | CRC only | CRC only |
| | aOR (95% CI) | aOR (95% CI) | aOR (95% CI) | aOR (95% CI) | aOR (95% CI) | aOR (95% CI) |
| Depression | *1.35 (1.11, 1.63)* | 1.05 (0.85, 1.29) | *1.47 (1.09, 1.99)* | *0.78 (0.68, 0.89)* | 1.09 (0.85, 1.41) | *1.41 (1.07, 1.84)* |
| Skin cancer | 1.04 (0.73, 1.48) | 0.78 (0.53, 1.15) | 0.75 (0.48, 1.17) | *0.75 (0.61, 0.92)* | *0.72 (0.54, 0.97)* | 0.97 (0.69, 1.36) |
| Other cancer | *2.90 (2.09, 4.05)* | *2.28 (1.57, 3.31)* | *3.08 (1.94, 4.91)* | *0.78 (0.62, 0.99)* | 1.06 (0.75, 1.51) | 1.35 (0.90, 2.03) |

aOR, adjusted odds ratio; CI, confidence interval; CRC, colorectal cancer. Bold values = statistically significant.

*'Other means American Indian, Native Hawaiian, Asian, Multiracial, and 'Other'.*

†Persons may have more than one condition or cancer.

women with a past history of diabetes (aOR = **1.79 (1.25, 2.55)**) or depression (aOR = *1.47 (1.09, 1.99)*) were more likely to screen for **CRC only** compared to neither screening. Conversely, women who were American Indian, Native Hawaiian, Asian, or multiracial were significantly less likely to be up-to-date for CRC screening compared to neither screening (aOR = **0.37 (0.21, 0.64)**).

### No or single screen compared to dual-screening

2a. Only Hispanic women compared to White women had significantly higher adjusted odds of having **cervical cancer screening alone** compared to dual-screening (aOR = *1.39 (1.10, 1.77)*). Alternatively, older women (aOR = *0.44 (0.39, 0.51)* for 55–59 years old and aOR = *0.31 (0.27, 0.36)* for 60–64 years old), Black women compared to White (aOR = *0.77 (0.64, 0.92)*), that graduated college compared to less than high school (aOR = *0.60 (0.46, 0.79)*), those with higher incomes (aOR = *0.66 (0.58, 0.75)*), or those with arthritis (aOR = *0.70 (0.61, 0.80)*), depression (aOR = *0.78 (0.68, 0.89)*), skin cancer (aOR = *0.75 (0.61, 0.92)*), or other cancers (aOR = *0.78 (0.62, 0.99)*) were significantly less likely to have **cervical cancer screening alone** compared to completing both screens.

2b. Only women 60–64 years old compared to younger women had significantly higher adjusted odds for having **CRC only screening** compared to both screens (aOR = *1.75 (1.30, 2.35)*). Women who were American Indian, Native Hawaiian, Asian, or Multiracial (aOR = *0.51 (0.33, 0.80)*), graduated from college (aOR = *0.50 (0.31, 0.81)*), had a higher income (aOR = *0.64 (0.49, 0.84)*), or had past history of skin cancer (aOR = *0.72 (0.54, 0.97)*) were significantly less likely to be up-to-date for **CRC screening alone** compared to both screens.

### Only CRC screening compared to only cervical cancer screening

3a. Older women compared to younger women had the highest adjusted odds of completing only the CRC screening compared to cervical cancer screening alone (aOR = *2.93 (2.17, 3.96)*) for 55–59 years old and aOR = *5.63 (4.14, 7.66)* for 60–64 years old. In addition, women with a past history of arthritis (aOR = *1.70 (1.31, 2.19)*) or depression (aOR = *1.41 (1.07, 1.84)*) had significantly higher adjusted odds of having only CRC screening compared to only cervical cancer screening. Women who were American Indian, Native Hawaiian, Asian, or Multiracial (aOR = *0.49 (0.30, 0.80)*) were less likely to have only CRC screening compared to only cervical cancer screening.

## Discussion

### Screening rates low in the woman-centered analysis

BRFSS' large-scale nationally weighted survey data show a new perspective that the rate of women who screened for both cervix cancer and CRC (58.2%) is far below the recommended rates for each cancer individually (HP2030 goals cervix: 84.3% and CRC: 74.4%) (*US Department of Health & Human Services, 2013*), HP2030-C07 (*Harper et al., 2021a*). Another new finding is that women 50–64 years old screen for cervical cancer alone five times more often than CRC alone. We show that the 50- to 64-year age range is dynamic with changing cancer screening behaviors from predominantly cervical screening at 50 years to mostly CRC screening at 65 years. These results are similar to the population survey study completed in southeast Michigan (*Harper et al., 2021a*).

### Screening decision making at multiple levels driven by age of the woman

2a. Physician level. Age becomes the most interesting factor predicting the divergence of the two cancer screening rates. One hypothesis is that it may be due to the specialty of the physician to whom the woman visits, a gynecologist vs. a primary care physician (PCP; family physician vs. general internist vs. geriatrician). A study of mammography orders noted that gynecologists provided care for 15% of the population who had mammograms ordered for ages 50–75 years, a small percentage of the age-appropriate screening population (*Taplin et al., 1994*). In addition, there is a 90% drop in visits to a gynecologist for prevention-related visits for those who are 45 years or older compared to the 18- to 44-year-old group, with a concomitant increase in PCP visits during this age transition (*Scholle et al., 2002*). This indicates that the **decision level** for the type of cancer screening could be at the *physician level* where specialty may influence the cancer screening (cervical vs. colorectal vs. both) offered to the woman.

2b. Patient level. Another **decision level** for screening could be at the *level of the woman*. The effects of screening by the age divergence could be due to physiologic aging with menopause at an average of 52 years in the United States (*US Department of Health & Human Services, 2021*; *Green and Santoro, 2009*). She may choose not to have any further cervical cancer screenings by vaginal speculum examination due to a variety of possible reasons: painful examinations, sexually inactive, no symptoms, or other personal reasons including past abuse (*Cadman et al., 2012*; *Saunders et al., 2021*; *Güneş and Karaçam, 2017*). She may choose to wait until 65 years to start CRC screening because this is the time that the Welcome to Medicare and Annual Medicare Wellness Visit overtly discuss CRC screening.

2c. Health system level. Decision making about screening could also reside in *the health system* as was evident during COVID when mailed FIT tests were sent to enrollees in response to the drop in cancer screenings (*BlueCross Blue Shield of Texas, 2020*; *Gupta et al., 2020*; *Gorin et al., 2021*; *Van Hoy, 2020*). It is well established that CRC screening, as recommended by the USPSTF (*Force et al., 2016*; *Force et al., 2021*) can be completed by any of six available tests, three of which are home based: FOBT, FIT, and multitarget stool DNA (FIT-DNA, Cologuard). Most recently, the American Cancer Society (ACS) and the USPSTF have recommended primary HPV testing for women 25/30–65 years for cervical cancer screening (*Fontham et al., 2020*; *Curry et al., 2018*) as the test that provides the most benefit for the least harm which can also be an at-home test, a well-accepted option (*Maver and Poljak, 2020*; *Kim et al., 2018*; *El Khoury et al., 2021* ). Nevertheless, our data show that of those women who only choose one cancer to screen for, cervical cancer will be almost five times more likely to be the screen chosen. We hypothesize from our study that If a home-based cervical cancer screening was FDA approved, it could enhance uptake of the already FDA approved CRC home-based testing for women in this vulnerable age range (*Bakr et al., 2020*; *Jaklevic, 2020*).

Another hypothesis-generating result of this study is that the high rates of a single cancer screen among the least educated may represent screening divorced from their routine primary care. Perhaps the single screen was completed at a local health fair, worksite, or a targeted program for that single cancer (*NBCCEDP, 2020*; *CRCCP, 2020*), achieving a maximum screening rate for this cancer, still lagging the HP2030 goals. We have shown that women who screen for one cancer do not necessarily screen for other cancers. In prior work, we have shown that having a strong relationship with a PCP leads to greater participation in dual cervical cancer and CRC screens (*Harper et al., 2021a*; *Harper et al., 2021b*) and we encourage leveraging the power of the PCP–woman dyad to address her multiple competing health needs. We showed that the single screen predictors of self-reported location (urban vs. rural) and employment status (*Harper et al., 2020*) were not associated with dual-screening in the final multivariate multinominal model. Again, this could support the hypothesis that the relationship between the PCP and the woman rather than her location or her employment status may increase cancer screening rates beyond the single cancer screen.

## Limitations

BRFSS is a highly respected and influential database for the US healthcare system reporting chronic health conditions and the use of preventive services. It is also a cross-sectional self-report survey whose responses are not equivalent to validated medical records (*St Clair et al., 2017*). Others have shown that the reported cancer screens could be mistakenly overestimated by about half of those who did not have a screen, whereas they accurately reflect those who have screened (*Anderson et al., 2019*; *Bonafede et al., 2019*). It is essential to point out, in support of using BRFSS, though, that BRFSS collects data in all 50 states, the District of Columbia, and three US territories interviewing more than 400,000 adults each year, making it the largest continuously conducted health survey system in the world. Nearly two-thirds of the US states use BFSS data to support health-related legislative efforts.

Using insurance claims data as the source of data extraction provide another option for analysis. Large populations with longitudinal follow-up and specific billing codes for cancer screenings are advantages. Still, the proven disadvantages include only having an insured population represented, incomplete billing data for each patient encounter, inaccurate coding data, and missing data altogether (*Stein et al., 2014*).

16.6% of the potential analysis cohort did not answer at least one of the cervical cancer or CRC screening questions. This nonresponse bias was considered by the inverse propensity scoring for missingness as detailed in Methods.

Even with accurate reporting by women themselves, a limitation of our work is that women were likely not aware of the screening modality used, and hence overestimated Pap alone screening when cotesting was used (*Watson et al., 2018*). For women 50–65 years old, though, cotesting is less common than Pap testing alone, mitigating that possible overestimation (*Watson et al., 2018*). In addition, cervical cancer screening modalities in the United States at the time of the survey were 80% cotesting and 20% cytology alone, indicating 14% of the women responded inaccurately to the type of cervical cancer screen they had (*Cuzick et al., 2021*).

Our age range, 50–64 years, was aligned with the screening guidelines at the time of the BRFSS 2018 survey. The most recent CRC recommendations by USPSTF move the age of initiation of CRC screening to 45 years, providing a greater potential age gap to be addressed for dual-screening. Finally, our work applies to the United States and the healthcare structure it provides. These results may not apply to other national healthcare systems.

## Conclusions

We showed that the risk factors associated with dual cancer screening are different from those associated with single cancer screens and that the risk factors for cervical cancer screening only are different from CRC screening only.

## Additional information

### Competing interests

Diane M Harper: Deputy Editor, *eLife*. The other authors declare that no competing interests exist.

### Funding

| Funder | Grant reference number | Author |
| --- | --- | --- |
| NCATS | UL1TR002240 | Diane M Harper<br>Ananda Sen |
| NCI | P30CA046592 | Diane M Harper<br>Ananda Sen |

The funders had no role in study design, data collection, and interpretation, or the decision to submit the work for publication.

### Author contributions

Diane M Harper, Conceptualization, Formal analysis, Funding acquisition, Investigation, Methodology, Writing – original draft; Melissa Plegue, Formal analysis, Methodology, Software, Validation, Writing – review and editing; Masahito Jimbo, Sherri Sheinfeld Gorin, Writing – review and editing; Ananda Sen, Formal analysis, Methodology, Writing – review and editing

### Author ORCIDs

Diane M Harper ⬚ http://orcid.org/0000-0001-7648-883X

### Ethics

Exempt IRB.

### Decision letter and Author response

Decision letter https://doi.org/10.7554/eLife.76070.sa1
Author response https://doi.org/10.7554/eLife.76070.sa2

## Additional files

### Supplementary files
- MDAR checklist
- Reporting standard 1. STROBE statement.

## Data availability

The BRFSS database is freely available online at https://www.cdc.gov/brfss/annual_data/annual_2018.html.

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

## Appendix 1

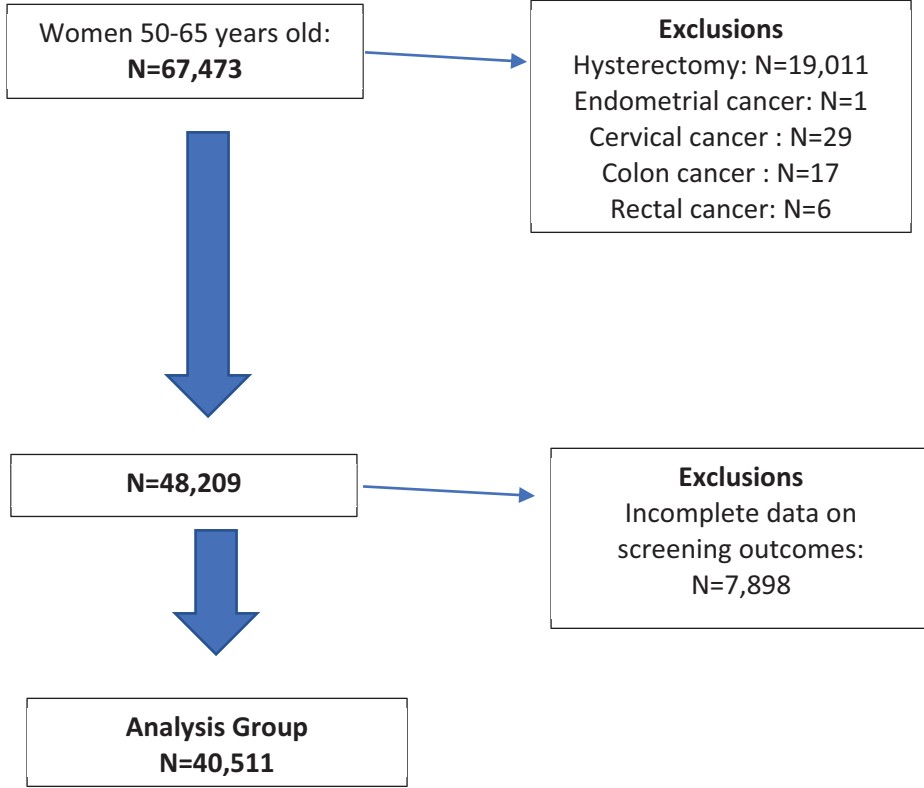

**Appendix 1—figure 1.** Consort diagram.

## Appendix results

*Appendix 1—table 1* presents the unadjusted multinomial regressions for the other comparisons within the multinomial models.

**Appendix 1—table 1.** Unadjusted multinomial regression.

| | Referent outcome: neither screen | | | Referent outcome: dual screens | | Referent outcome: cervical only |
|---|---|---|---|---|---|---|
| | Dual screens | Cervical only | CRC only | Cervical only | CRC only | CRC only |
| | OR (95% CI) | OR (95% CI) | OR (95% CI) | OR (95% CI) | OR (95% CI) | OR (95% CI) |
| **Age** | | | | | | |
| 50–54 | *Ref* | *Ref* | *Ref* | *Ref* | *Ref* | *Ref* |
| 55–59 | *1.29 (1.07, 1.56)* | *0.55 (0.45,0.68)* | *1.68 (1.24, 2.29)* | *0.43 (0.37, 0.49)* | *1.30 (1.01, 1.69)* | *3.05 (2.31, 4.01)* |
| 60–64 | *1.48 (1.24, 1.78)* | *0.45 (0.37, 0.56)* | *2.74 (2.04, 3.70)* | *0.31 (0.27, 0.35)* | *1.85 (1.44, 2.39)* | *6.04 (4.60, 7.93)* |
| **Race** | | | | | | |
| White | *Ref* | *Ref* | *Ref* | *Ref* | *Ref* | *Ref* |
| Black | *1.28 (0.98, 1.67)* | *1.33 (1.01, 1.76)* | 1.20 (0.77, 1.86) | 1.04 (0.88, 1.22) | 0.93 (0.64, 1.36) | 0.90 (0.61, 1.33) |
| Other* | 0.73 (0.49, 1.09) | 0.76 (0.49, 1.17) | *0.35 (0.21, 0.60)* | 1.04 (0.77, 1.40) | *0.49 (0.32,0.74)* | *0.47 (0.29, 0.74)* |
| Hispanic | 0.80 (0.64, 1.00) | *1.63 (1.25, 2.12)* | 0.96 (0.62, 1.51) | *2.04 (1.67, 2.50)* | 1.21 (0.80,1.83) | *0.59 (0.38, 0.91)* |
| **Education** | | | | | | |
| Less than high school | *Ref* | *Ref* | *Ref* | *Ref* | *Ref* | *Ref* |
| High school graduate | *1.54 (1.20, 1.98)* | 0.90 (0.67, 1.21) | 1.14 (0.76, 1.72) | *0.58 (0.46, 0.75)* | 0.74 (0.51, 1.08) | 1.27 (0.84, 1.91) |

*Appendix 1—table 1 Continued on next page*

*Appendix 1—table 1 Continued*

| | Referent outcome: neither screen | | | Referent outcome: dual screens | | Referent outcome: cervical only |
|---|---|---|---|---|---|---|
| | Dual screens | Cervical only | CRC only | Cervical only | CRC only | CRC only |
| | OR (95% CI) | OR (95% CI) | OR (95% CI) | OR (95% CI) | OR (95% CI) | OR (95% CI) |
| Attended college/Tech school | *2.62 (2.03, 3.37)* | 1.29 (0.96, 1.74) | 1.42 (0.96, 2.11) | *0.49 (0.39, 0.63)* | *0.54 (0.38, 0.78)* | 1.10 (0.75, 1.62) |
| Graduated college/Tech school | *4.84 (3.67, 6.38)* | *1.96 (1.43, 2.69)* | *1.64 (1.09, 2.45)* | *0.41 (0.32, 0.52)* | *0.34 (0.24, 0.48)* | 0.83 (0.57, 1.22) |
| **Income** | | | | | | |
| <$50K | Ref | Ref | Ref | Ref | Ref | Ref |
| >$50K | *3.30 (2.79, 3.90)* | *2.07 (1.72, 2.48)* | *1.60 (1.24, 2.07)* | *0.63 (0.56, 0.70)* | *0.49 (0.39, 0.60)* | *0.78 (0.62, 0.97)* |
| **Marital status** | | | | | | |
| Married or partnered | Ref | Ref | Ref | Ref | Ref | Ref |
| Single[†] | *0.56 (0.48, 0.66)* | *0.68 (0.57,0.81)* | *0.78 (0.61, 0.99)* | *1.21 (1.08, 1.35)* | *1.38 (1.13,1.69)* | 1.14 (0.92, 1.42) |
| **Occupational status** | | | | | | |
| Employed | Ref | Ref | Ref | Ref | Ref | Ref |
| Unemployed, looking[‡] | *0.56 (0.41, 0.78)* | *0.64 (0.46, 0.88)* | 0.81 (0.51, 1.28) | 1.12 (0.87, 1.45) | 1.43 (0.95, 2.17) | 1.27 (0.83, 1.94) |
| Unemployed, not looking[‡] | 0.99 (0.83, 1.19) | *0.76 (0.60, 0.96)* | 1.22 (0.92, 1.63) | *0.76 (0.64, 0.91)* | 1.23 (0.97, 1.57) | *1.61 (1.22, 2.13)* |
| Unable to work | *0.64 (0.52,0.79)* | *0.64 (0.50, 0.81)* | *1.52 (1.10, 2.10)* | 0.99 (0.84, 1.16) | *2.36 (1.81, 3.08)* | *2.39 (1.78, 3.20)* |
| **Location** | | | | | | |
| Urban | Ref | Ref | Ref | Ref | Ref | Ref |
| Rural | *0.64 (0.52, 0.78)* | *0.76 (0.61, 0.95)* | 0.97 (0.72, 1.30) | *1.20 (1.04, 1.39)* | *1.52 (1.20, 1.94)* | 1.27 (0.98, 1.65) |
| **Chronic conditions[§]** | | | | | | |
| Cardiac | *0.64 (0.48, 0.87)* | *0.59 (0.42,0.83)* | 1.11 (0.74, 1.65) | 0.92 (0.74, 1.14) | *1.72 (1.26, 2.35)* | *1.87 (1.33, 2.64)* |
| Stroke | 0.96 (0.64, 1.42) | 0.80 (0.52, 1.23) | 0.90 (0.54, 1.51) | 0.84 (0.64, 1.10) | 0.94 (0.63, 1.39) | 1.12 (0.73, 1.72) |
| Lung | 0.95 (0.80, 1.14) | 0.89 (0.70, 1.13) | 1.15 (0.87, 1.53) | 0.93 (0.77, 1.12) | 1.21 (0.95, 1.54) | 1.30 (0.97, 1.74) |
| Arthritis | *1.51 (1.30, 1.76)* | 0.99 (0.83, 1.18) | *2.27 (1.80, 2.86)* | *0.65 (0.58, 0.74)* | *1.50 (1.24, 1.81)* | *2.29 (1.85, 2.82)* |
| Kidney | 0.84 (0.57, 1.23) | *0.56 (0.36, 0.87)* | 0.76 (0.44, 1.30) | *0.66 (0.49, 0.91)* | 0.90 (0.59, 1.39) | 1.36 (0.83, 2.23) |
| Diabetes | 1.04 (0.83, 1.29) | *1.18 (0.89, 1.58)* | *1.72 (1.24, 2.38)* | *1.66 (1.27, 2.15)* | *1.66 (1.27, 2.15)* | *1.46 (1.05, 2.01)* |
| Depression | 1.16 (0.99, 1.37) | 0.97 (0.79, 1.20) | *1.56 (1.21, 2.01)* | *0.84 (0.72, 0.98)* | *1.34 (1.08, 1.65)* | *1.60 (1.25, 2.05)* |
| Skin cancer | *1.41 (1.01, 1.95)* | 0.80 (0.56, 1.14) | 1.11 (0.73, 1.67) | *0.57 (0.48, 0.68)* | 0.79 (0.60, 1.04) | *1.38 (1.01, 1.88)* |
| Other cancer | *3.03 (2.26, 4.06)* | *2.16 (1.54, 3.03)* | *3.14 (2.07, 4.77)* | *0.71 (0.58, 0.88)* | 1.04 (0.75, 1.43) | *1.45 (1.01, 2.09)* |

Significant results are shown in **bold italic** font.
[*]Other means American Indian, Native Hawaiian, Asian, Multiracial and 'Other'.
[†]Single includes divorced, widowed, separated, and never married.
[‡]'Unemployed, not looking' includes 'Homemaker, Student, and Retired', 'Unemployed, looking' includes out of work over a year and out of work less than a year.
[§]Persons may have more than one condition or cancer.

## Screening compared to neither screening

1a. Compared to neither screening, significant predictors of being up-to-date for **dual-screening** were older age, being Black compared to White, having more education, having more income, having arthritis, and having a past skin or any kind of cancer. Alternatively, those who were looking for employment, unable to work, living in a rural area, and having past cardiac disease were the least likely to have the dual screens.

1b. Compared to neither screening, significant predictors of being up-to-date for **only the cervical cancer screening** were women of Black or Hispanic races compared to White, having graduated from college, having more income and having a history of diabetes or another cancer. Older women, being single, unemployed for any reason and unable to work, living in a rural area, having past cardiac or kidney disease, were less likely to have only cervical cancer screening.

1c. Compared to neither screening, significant predictors of being up-to-date for **only CRC screening** were older age, having more education, having more income, and having had arthritis,

diabetes, depression, or other cancers. Those least likely to have only CRC screening were single women and those unemployed but looking for a job.

## No or some screening compared to dual-screening

2a. Compared to having dual-screening, significant predictors of only cervical cancer screening were women who were Hispanic compared to White, single, living in rural area or had a past history of diabetes. Women who were older, had higher education, higher income, were unemployed, and had a past history of arthritis, kidney disease, depression, skin, and other cancers were less likely to screen for **cervical cancer alone**.

2b. Furthermore, compared to having dual-screening, significant predictors of **a CRC screen alone** were women who were older, single, unable to work, lived in a rural area, had past cardiac, arthritis, diabetes, or depression. Factors that make CRC screening alone, compared to being up-to-date for both screens less likely were being either American Indian, Native Hawaiian, Asian, or multiracial, having higher education, and higher income.

## CRC screening alone compared to cervical cancer screening alone

3a. Women who were older, unemployed, unable to work, had a past history of cardiac, arthritis, diabetes, depression, skin cancer, or other cancers were significantly more likely to have a **single CRC cancer screen compared to a cervical cancer screen**.

3b. On the other hand, women who were Hispanic, American Indian, Native Hawaiian, Asian, or multiracial compared to White, and had a higher income were more likely to have a single cervical cancer screen rather than a CRC screen

