## [Editor Report]

This work presents a different US perspective for cancer screenings among the least screened women, those 50-64 years old. Using the Behavioral Risk Factor Surveillance System and the two cancer screenings (cervical and colorectal) done by her primary care physician, the findings of this work show that women-centric screening is lower than single cancer screenings. There is a large gap in health systems processes to meet the cancer screening needs of each woman.

---

## [Decision Letter]

**Decision letter after peer review:**

Thank you for submitting your article "US women screen at low rates for both cervical and colorectal cancers than a single cancer: a cross-sectional population-based observational study" for consideration by *eLife*. Your article has been reviewed by 3 peer reviewers, and the evaluation has been overseen by a Reviewing Editor and a Senior Editor. The following individuals involved in review of your submission have agreed to reveal their identity: Tatiana Ramirez (Reviewer #1).

As is customary in *eLife*, the reviewers have discussed their critiques with one another. What follows below is the Reviewing Editor's edited compilation of the essential and ancillary points provided by reviewers in their critiques and in their interaction post-review. Please submit a revised version that addresses these concerns directly. Although we expect that you will address these comments in your response letter, we also need to see the corresponding revision clearly marked in the text of the manuscript. Some of the reviewers' comments may seem to be simple queries or challenges that do not prompt revisions to the text. Please keep in mind, however, that readers may have the same perspective as the reviewers. Therefore, it is essential that you attempt to amend or expand the text to clarify the narrative accordingly.

Essential revisions:

The reviewers agreed that this manuscript addresses an important question and that the results were worthy of consideration for publication. However, the reviewers felt that the limitations of relying on self-report data from the Behavioral Risk Factor Surveillance System should be given careful attention, and that the scope of the manuscript should be narrowed and concluding statements tempered. The reviewers provided editorial comments below that should also be addressed.

*Reviewer #1 (Recommendations for the authors):*

It is an interesting paper with potential to be accepted in *eLife*. In general, I find the manuscript correctly written.

*Reviewer #2 (Recommendations for the authors):*

The authors are investigating an important question. However, this paper needs major editing to convey clear, concise and appropriate information. As detailed in the public sections, there is inconsistency in terms used – dual screening vs by-woman analysis. In addition you should refer to screening as the women reported being UTD with X screening. This is BRFSS data – women did not choice to screen – they reported screening – you have no Medical Record Data so you have to be very careful how you phrase things. It is not clear why you are focusing only on cervical and CRC screening.

To reduce the confusion in the presentation of the results, I suggest you tighten that up – too many comparisons that seem repetitive. You seem to put too much weight on descriptive and unadjusted results – why not put the unadjusted results in a shorter section after the main results. The numbers on your Graphical Abstract don't add up – and this is not a "New" way – it's your proposed way or the results of this analysis.

Your conclusions need to be tempered by what your data shows – don't over stretch – and this is self-reported data from 2018. You make claims that are not supported – e.g. comparing HP 2030 goals that are derived by single screen to your dual measure – that is not a fair comparison. Lastly, there are some published studies that have examined multiple screening behaviors in terms of interventions. You might want to include them.

*Reviewer #3 (Recommendations for the authors):*

1) Further discussion over the reason for a significant value for geographic location should be discussed- is this related to access?

2) The authors discuss physician decision level- how can this be gleaned through these data? A provider can make a recommendation for a CRC and even book the GI appt- but a patient actually has to show up.

3) We know screening is not necessarily linked to primary care and, particularly cervical cancer screening, happens in a silo as many women who have not gotten a Pap or HPV test in the US has seen a provider. How do these data clarify this- if so, needs to be further discussed in discussion.

4) Self reported data and claims data are very different. Often for cervical cancer screening, a patient might have thought a speculum exam is a Pap or HPV. When it is not. How would this affect these data- assume under-reporting of Pap, which should be stated.

---

## [Author Response]

Essential revisions:The reviewers agreed that this manuscript addresses an important question and that the results were worthy of consideration for publication. However, the reviewers felt that the limitations of relying on self-report data from the Behavioral Risk Factor Surveillance System should be given careful attention, and that the scope of the manuscript should be narrowed and concluding statements tempered. The reviewers provided editorial comments below that should also be addressed.

We appreciate your concern about a self-report survey. We want to call to your attention that BRFSS is the United States' premier system of health-related telephone surveys that collect state data about US residents regarding their health-related risk behaviors, chronic health conditions, and use of preventive services. BRFSS collects data in all 50 states, the District of Columbia, and three US territories. BRFSS completes more than 400,000 adult interviews each year, making it the largest continuously conducted health survey system globally. US Centers for Disease Control and Prevention (CDC) sponsors the BRFSS annual survey.

BRFSS is a highly respected and influential database for US health care policy. BRFSS data help each state to establish and track state and local health objectives, plan health programs, implement disease prevention and health promotion activities, and monitor trends. Nearly two-thirds of states use BRFSS data to support health-related legislative efforts.

Many authors have extensively studied the concern about inaccurate self-reporting, with citation 37 (Anderson 2019) definitively showing that those who are screened accurately report this in a self-report survey. Those who did not complete the screen will inaccurately report by about 50% that they did screen, meaning that BRFSS results are potentially overestimated. This realization makes our results and conclusions more potent as we are most likely not even at our reported rates for dual screening.

To honor the concern about the scope of the manuscript and its conclusions, we have added to the limitations section of the Discussion section the following:

“It is essential to point out, in support of using BRFSS, though, that BRFSS collects data in all 50 states, the District of Columbia, and three US territories interviewing more than 400,000 adults each year, making it the largest continuously conducted health survey system in the world. Nearly two-thirds of the US states use BFSS data to support health-related legislative efforts.

Using insurance claims data as the source of data extraction provides another option for analysis. Large populations with longitudinal follow-up and specific billing codes for cancer screenings are advantages. Still, the proven disadvantages include only having an insured population represented, incomplete billing data for each patient encounter, inaccurate coding data, and missing data altogether (Stein 2015).”

Reviewer #2 (Recommendations for the authors):The authors are investigating an important question. However, this paper needs major editing to convey clear, concise and appropriate information. As detailed in the public sections, there is inconsistency in terms used – dual screening vs by-woman analysis. In addition you should refer to screening as the women reported being UTD with X screening.

Dual-screening refers to having both screens up to date in the same person. We have changed the terminology in the graphic abstract to reflect woman-centered screening instead of population-centered screening, which has been the traditional epidemiologic unit of measurement.

We state in the Methods section how women are considered up to date with each screening modality for each cancer.

This is BRFSS data – women did not choice to screen – they reported screening – you have no Medical Record Data so you have to be very careful how you phrase things. It is not clear why you are focusing only on cervical and CRC screening.

We agree with the reviewer. Women reported their perceptions of screening to the BRFSS survey.

We are focusing on cervical and colorectal cancer screening because both of these cancers can be efficiently screened at home by the person for the average risk woman.

To reduce the confusion in the presentation of the results, I suggest you tighten that up – too many comparisons that seem repetitive. You seem to put too much weight on descriptive and unadjusted results – why not put the unadjusted results in a shorter section after the main results.

We specifically chose to do the multinomial multivariate analyses because the information we were seeking could only be answered in this way.

As you are aware, multivariate, multinomial logistic regression requires several interpretation paragraphs because each comparison answers a different question. To tighten the results, we have moved the translation of the supplementary table 1 results to the appendix.

The numbers on your Graphical Abstract don't add up – and this is not a "New" way – it's your proposed way or the results of this analysis.

With respect, the total is 100%.

9%+27%+589%+5%=100% for women-centered perspective

The traditional population-based epi metric uses 100% as the total population screened for that cancer.

This work offers a new perspective on cancer screening. To date, the public health reporting has only been on what percent of the population has received a single cancer screen. We have made our perspective patient-oriented. The new way forward is to consider each person's completion of the cancer screening recommendations that she can do and eventually these will be at home.

Your conclusions need to be tempered by what your data shows – don't over stretch – and this is self-reported data from 2018. You make claims that are not supported – e.g. comparing HP 2030 goals that are derived by single screen to your dual measure – that is not a fair comparison.

With respect, the BRFSS data are used for Healthy People goals by definition. (Song S, White A, Kucik JE. Use of Selected Recommended Clinical Preventive Services – Behavioral Risk Factor Surveillance System, United States, 2018. MMWR Morb Mortal Wkly Rep. 2021 Apr 2;70(13):461-466. doi: 10.15585/mmwr.mm7013a1. PMID: 33793461; PMCID: PMC8022875.) The yearly BRFSS rates are compared to Healthy People goals (Zimmerman FJ, Anderson NW. Trends in Health Equity in the United States by Race/Ethnicity, Sex, and Income, 1993-2017. JAMA Netw Open. 2019 Jun 5;2(6):e196386. doi: 10.1001/jamanetworkopen.2019.6386. Erratum in: JAMA Netw Open. 2019 Jul 3;2(7):e199357. PMID: 31251377; PMCID: PMC6604079.)

Lastly, there are some published studies that have examined multiple screening behaviors in terms of interventions. You might want to include them.

With respect, we could not find any other paper that was woman-centric in screening tests. We did find several papers that evaluated several cancer screening tests from the population-centric perspective.

Reviewer #3 (Recommendations for the authors):1) Further discussion over the reason for a significant value for geographic location should be discussed- is this related to access?

The geographic location's significance was in the univariate multinomial regression analyses presented in the supplementary table. Because these significances did not remain once the multivariate, multinomial regression was completed, geographic location has no bearing on the final outcome that compares dual-screening to a single screen.

2) The authors discuss physician decision level- how can this be gleaned through these data? A provider can make a recommendation for a CRC and even book the GI appt- but a patient actually has to show up.

As discussed above, we hypothesize, based on age being a significant covariate in our results, that the age of the woman is often linked to the type of physician she has for her care. For instance, a woman with a family physician could have both cancer screenings done or ordered in the same visit. The older she gets, the less likely she will have a physician who does cervical cancer screening anymore (e.g., a general internist or geriatrician). We explain why age has such a different type of significance for the two screenings.

We agree that whether the patient does their cancer screening at home or comes to the GI clinic for colonoscopy is a behavior that is often mediated by the type of relationship the patient has with her ordering PCP doctor.

3) We know screening is not necessarily linked to primary care and, particularly cervical cancer screening, happens in a silo as many women who have not gotten a Pap or HPV test in the US has seen a provider. How do these data clarify this- if so, needs to be further discussed in discussion.

We appreciate the reviewer's comment. From a woman-centric perspective, though, we are the first to show that getting both screens is not nearly as common as getting only cervical cancer screening.

We hypothesize that single cancer screening is more common from a population perspective because programs offer cervical cancer screening in the community. No one asks the woman at her church fair for cervical cancer screening if she is also getting her colorectal cancer screening.

4) Self reported data and claims data are very different. Often for cervical cancer screening, a patient might have thought a speculum exam is a Pap or HPV. When it is not. How would this affect these data- assume under-reporting of Pap, which should be stated.

We agree with the reviewer and have discussed this above.